# Learning clinical networks from medical records based on information estimates in mixed-type data

**Vincent Cabeli**[1,2☯], **Louis Verny**[1,2☯], **Nadir Sella**[1,2,3☯], **Guido Uguzzoni**[1,2], **Marc Verny**[2,4]*, **Hervé Isambert**[1,2]*

**1** Institut Curie, PSL Research University, CNRS, UMR168, 26 rue d'Ulm, 75005 Paris, France, **2** Sorbonne Université, 4, place Jussieu, 75005 Paris, France, **3** LIMICS, UMRS 1142, 15 rue de l'école de médecine, 75006 Paris, France, **4** Hôpital La Pitié-Salpêtrière, 47-83 boulevard de l'Hôpital, 75013 Paris, France

☯ These authors contributed equally to this work.
* marc.verny@aphp.fr (MV); herve.isambert@curie.fr (HI)

## Abstract

The precise diagnostics of complex diseases require to integrate a large amount of information from heterogeneous clinical and biomedical data, whose direct and indirect interdependences are notoriously difficult to assess. To this end, we propose an efficient computational approach to simultaneously compute and assess the significance of multivariate information between any combination of mixed-type (continuous/categorical) variables. The method is then used to uncover direct, indirect and possibly causal relationships between mixed-type data from medical records, by extending a recent machine learning method to reconstruct graphical models beyond simple categorical datasets. The method is shown to outperform existing tools on benchmark mixed-type datasets, before being applied to analyze the medical records of eldery patients with cognitive disorders from La Pitié-Salpêtrière Hospital, Paris. The resulting clinical network visually captures the global interdependences in these medical records and some facets of clinical diagnosis practice, without specific hypothesis nor prior knowledge on any clinically relevant information. In particular, it provides some physiological insights linking the consequence of cerebrovascular accidents to the atrophy of important brain structures associated to cognitive impairment.

## Author summary

We developed a machine learning approach to analyze medical records and help clinicians visualize the direct and indirect interrelations between clinical examinations and the variety of syndromes implicated in complex diseases. The reconstruction of such clinical networks is illustrated on the spectrum of cognitive disorders, originating from either neurodegenerative, cerebrovascular or psychiatric dementias. This global network analysis is also shown to uncover novel direct associations and possible cause-effect relationships between clinically relevant information, such as medical examinations, diagnoses, treatments and personal data from patients' medical records.

**Data Availability Statement:** All relevant data, properly de-identified to guarantee the anonymity of patients, is provided as a supplementary table (S1 Table).

**Funding:** HI received funding from IRIS data science program of PSL university, DIM program from Region Ile-de-France and Labex celtisphybio. The funders had no role in study design, data collection and analysis, decision to publish, or preparation of the manuscript.

## Introduction

The precise diagnostics of neurological disorders require to integrate a large amount of information from a variety of biomedical tests and clinical examinations. These diagnostics must also take into account age-related comorbid medical conditions, such as diabetes and cardiovascular diseases, which concern a large fraction of patients, as the incidence of neurodegenerative diseases increases with age. Such comorbid medical conditions influence neuropathology treatment decisions as well as short- and long-term survival of patients but are often overlooked in clinical trials. This situation underlines the need to directly analyze real life medical records to learn *clinical networks*, that are graphical models highlighting direct, indirect and possibly causal associations between clinically relevant information in patients' medical records.

Medical records contain, however, mixed types of data from simple binary or nominal variables (*i.e.*, with multiple unordered categories) to ordinal (*e.g.* neuropsychological test scales) or continuous (*e.g.* age, body mass index) variables, whose interdependences are not readily assessed within a unified information-theoretic framework. As mutual information is primarily defined between nominal variables, its estimation for continuous or mixed-type variables is notoriously difficult beyond the gaussian approximation of continuous distributions, for which a simple relation exists with correlation coefficients [1]. In particular, arbitrary discretization of continuous variables tends to underestimate mutual information for small number of bins, while overestimating it for large number of bins due to finite numbers of patients, as sketched in Fig 1. Moreover, so far, no rationale provides optimum bin partitions to estimate mutual information, for typical cohort size of patients. Alternatively, local metric approaches have been proposed to estimate mutual information [2] and conditional information [3–5], including between mixed-type variables [6–8], based on k-nearest neighbor (kNN) statistics. However, the statistical significance of kNN information estimates remains difficult to assess in practice [2, 9], thereby limiting their use to uncover (conditional) independences between continuous or mixed-type variables from real-life datasets.

In this paper, we first develop and implement an optimum binning method to simultaneously compute and assess the significance of mutual information, as well as conditional multivariate information, between any combination of continuous or mixed-type variables. The method is based on minimum description length principles [10, 11] and finds optimum bin

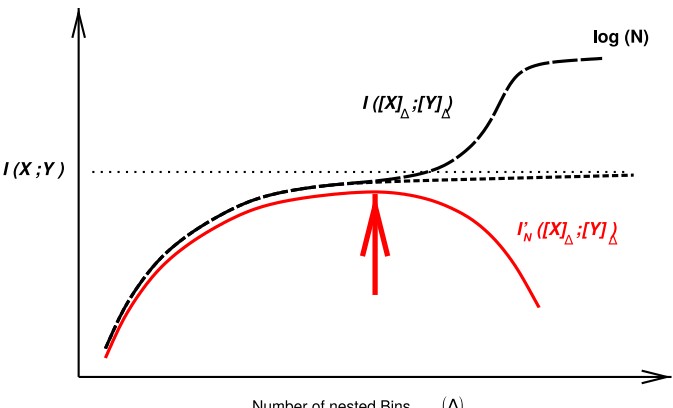

**Fig 1. Mutual information computation between continuous or mixed-type variables.** Outline of mutual information computation between continuous or mixed-type variables for a finite dataset of *N* samples. Mutual information is estimated through an optimum partitioning of continuous variable(s) (solid red line and arrow) after introducing a complexity term to account for the finite size of the dataset, see main text.

partitions, iteratively for each continuous variable, through an efficient dynamic programming scheme with quadratic complexity, $\mathcal{O}(N^2)$, where $N$ is the number of patients in the dataset. This efficient approach is then used to assess direct *versus* indirectcause-effect relationships between mixed-type data from medical records, by extending a recent network learning method [12, 13] to recontruct graphical models beyond simple categorical datasets.

The method is shown to outperform existing tools on benchmark mixed-type datasets, before being applied to analyze the medical records of eldery patients with cognitive disorders from La Pitié-Salpêtrière Hospital, Paris. The resulting clinical network visually captures the global interdependences in these medical records and some facets of clinical diagnosis practice, without specific hypothesis nor prior knowledge on any clinically relevant information. The reconstructed clinical network recovers well known as well as novel direct and indirect relations between medically relevant variables. In particular, it provides some physiological insights linking the consequence of cerebrovascular accidents to the atrophy of important brain structures associated to cognitive impairment.

## Methods

### Assessing information in continuous or mixed-type data

**Information-maximizing discretization of continous data.**   While mutual information is usually defined as a discrete summation over nominal variables, *i.e.*, $I(X;Y) = \sum_{x,y} p_{x,y} \log(p_{x,y}/p_x p_y)$, its most general definition consists in taking the supremum over all finite partitions, $\mathcal{P}$ and $\mathcal{Q}$, of variables, $X$ and $Y$ [1],

$$I(X;Y) = \sup_{\mathcal{P},\mathcal{Q}} I([X]_{\mathcal{P}}; [Y]_{\mathcal{Q}}) \tag{1}$$

which can be applied to continuous or mixed-type variables. Moreover, by continuing to refine some initial partitions through the addition of further cut points for continuous variable(s), one finds a monotonically increasing sequence [1], $I([X]_{\mathcal{P}}; [Y]_{\mathcal{Q}})$, as depicted on Fig 1. In practice, however, Eq 1 cannot be used to estimate $I(X;Y)$ from an actual dataset with finite sample size, as the refinement of partitions eventually assigns each of the $N$ different samples into $N$ different bins. This leads to a shift of convergence towards $\log N$ instead of the theoretical limit, $I(X;Y)$, which requires an infinite amount of data (dotted line in Fig 1).

In this paper, we propose to adapt Eq 1 to account for the finite number of samples in actual datasets,

$$I'_N(X;Y) = \sup_{\mathcal{P},\mathcal{Q}} I'_N([X]_{\mathcal{P}}; [Y]_{\mathcal{Q}}) \tag{2}$$

by introducing a finite size correction to mutual information,

$$I'_N([X]_{\mathcal{P}}; [Y]_{\mathcal{Q}}) = I_N([X]_{\mathcal{P}}; [Y]_{\mathcal{Q}}) - k'_{\mathcal{P};\mathcal{Q}}(N) \frac{1}{N} \tag{3}$$

where $k'_{\mathcal{P};\mathcal{Q}}(N)$ corresponds to a complexity term introduced in [14, 15] to discriminate between variable dependence (for $I'_N([X]_{\mathcal{P}}; [Y]_{\mathcal{Q}}) > 0$) and variable independence (for $I'_N([X]_{\mathcal{P}}; [Y]_{\mathcal{Q}}) \leqslant 0$) given a finite dataset of size $N$. In the present context of finding an optimum discretization for continuous variables, this complexity term introduces a penalty which eventually outweights the information gain in refining bin partitions further, when there is not enough data to support such a refined model, as depicted on Fig 1.

For discrete variables, typical complexity terms correspond to the Bayesian Information Criterion (BIC), $k^{\text{BIC}}_{\mathcal{P};\mathcal{Q}}(N) = 1/2(r_x - 1)(r_y - 1) \log N$, where $r_x$ and $r_y$ are the number of bins

for $X$ and $Y$, or the $X$- and $Y$-Normalized Maximum Likelihood (NML) criteria [14–16], defined as,

$$k_{\mathcal{P};\mathcal{Q}}^{X-\mathrm{NML}}(N) = \sum_{y}^{r_y} \log \mathcal{C}_{n_y}^{r_x} - \log \mathcal{C}_{N}^{r_x} \tag{4}$$

$$k_{\mathcal{P};\mathcal{Q}}^{Y-\mathrm{NML}}(N) = \sum_{x}^{r_x} \log \mathcal{C}_{n_x}^{r_y} - \log \mathcal{C}_{N}^{r_y} \tag{5}$$

where $\mathcal{C}_{n_y}^{r_x}$ is the parametric complexity associated with the $y$th bin of variable $Y$ containing $n_y$ samples, and similarly for $\mathcal{C}_{n_x}^{r_y}$ with the $n_x$-size bin of variable $X$ in Eq 5.

Parametric complexities $\mathcal{C}_n^r$ are defined by summing a multinomial likelihood function over all possible partitions of $n$ data points into a maximum of $r$ bins as,

$$\mathcal{C}_n^r = \sum_{\substack{\ell_1+\ell_2+\cdots+\ell_r=n}}^{\ell_k \geqslant 0} \frac{n!}{\ell_1!\ell_2!\cdots\ell_r!} \prod_{k=1}^{r} \left(\frac{\ell_k}{n}\right)^{\ell_k} \tag{6}$$

which can in fact be computed recursively in linear-time [17]. For large $n$ and $r$, inherent to large datasets with continuous or mixed-type variables, we found that $\mathcal{C}_n^r$ computation can be made numerically stable by implementing the recursion on parametric complexity ratios $\mathcal{D}_n^r = \mathcal{C}_n^r/\mathcal{C}_n^{r-1}$ rather than parametric complexities themselves as,

$$\mathcal{D}_n^r = 1 + \frac{n}{(r-2)\mathcal{D}_n^{r-1}} \tag{7}$$

$$\log \mathcal{C}_n^r = \sum_{k=2}^{r} \log \mathcal{D}_n^k \tag{8}$$

for $r \geqslant 3$, with $\mathcal{C}_n^1 = 1$ and $\mathcal{C}_n^2 = \mathcal{D}_n^2$, which can be computed directly with the general formula, Eq 6, for $r = 2$,

$$\mathcal{C}_n^2 = \sum_{h=0}^{n} \binom{n}{h} \left(\frac{h}{n}\right)^h \left(\frac{n-h}{n}\right)^{n-h} \tag{9}$$

or its Szpankowski approximation for large $n$ (needed for $n > 1000$ in practice) [18–20],

$$\mathcal{C}_n^2 = \sqrt{\frac{n\pi}{2}} \left(1 + \frac{2}{3}\sqrt{\frac{2}{n\pi}} + \frac{1}{12n} + \mathcal{O}\left(\frac{1}{n^{3/2}}\right)\right) \tag{10}$$

$$\simeq \sqrt{\frac{n\pi}{2}}\exp\left(\sqrt{\frac{8}{9n\pi}} + \frac{3\pi-16}{36n\pi}\right) \tag{11}$$

For continuous variables, however, the variable categories are not given *a priori* and need to be specified and thus encoded in the model complexity within the frame of the Minimum Description Length (MDL) principle [11]. In absence of priors for any specific partition with $r$ bins, the model index should be encoded with a uniform distribution over all partitions with the same number of bins [11]. As there are $\binom{N-1}{r_x-1}$ ways to choose $r_x - 1$ out of $N - 1$ possible cut points, corresponding to a codelength of $\log \binom{N-1}{r_x-1}$ for a continuous variable $X$ (and

similarly for $Y$ if it is continuous), the model complexity associated with the partitioning of continuous or mixed-type variablesbecomes,

$$k'_{\mathcal{P};\mathcal{Q}}(N) = k_{\mathcal{P};\mathcal{Q}}(N) + \log\binom{N-1}{r_x - 1} + \log\binom{N-1}{r_y - 1} \tag{12}$$

with $\log\binom{N-1}{r-1} = (r-1)\,C_{N,r}$, where $C_{N,r}$ corresponds to theencoding cost associated to each of the $r-1$ cut points with $r = r_x$ or $r_y$.

While finding the supremum of $I'_N([X]_{\mathcal{P}}; [Y]_{\mathcal{Q}})$ over *all* possible partitions $\mathcal{P}$ and $\mathcal{Q}$ according to Eq 2 seems intractable, it can be computed rather efficiently in practice.

The approach is inspired by the computation of an MDL-optimal histogram for a single continuous variable [11], which can be done exactly in $\mathcal{O}(N^3)$ steps. As the approach cannot be generalized to more than one variable, we implemented a local optimization heuristics, which finds the optimum cut points for each continuous variable, iteratively, keeping the partitions of the other continuous variable(s) fixed. This enables to gain an order of magnitude in the optimization running time at each iteration, which scales as $\mathcal{O}(N^2)$, as detailed below.

In practice for two variables, we start from an initial (or optimized) $X$ partition with $r_x$ bins of various sizes and an estimate of the number of $Y$ bins, $r_y^\circ$. The sample-scaled mutual information with finite size correction, *i.e.*, $nI'_n(X; Y)$, is then optimized iteratively for $n = 1, \cdots, N$ samples, over all $Y$ partitions, through the following $\mathcal{O}(N^2)$ dynamic programming scheme, using Eq 4 as parametric complexity,

$$nI'_n(X; Y) = \max_{0 \leqslant j < n} \Big[ jI'_j(X; Y) + \sum_x^{r_x} n_{xy} \log n_{xy} - n_y \log n_y - \log \mathcal{C}^{r_x}_{n_y} - C_{N,r_y^\circ} \Big] \tag{13}$$

where the last added $Y$ bin, including $n_y = n - j$ samples distributed over the $r_x$ bins of $X$ (with $\sum_x^{r_x} n_{xy} = n_y$), comes with an independent mutual information contribution, $\sum_x^{r_x} n_{xy} \log n_{xy} - n_y \log n_y$, a parametric complexity, $\log \mathcal{C}^{r_x}_{n_y}$, and encoding cost, $C_{N,r_y^\circ}$. The initial condition for $j = 0$ in (13) is set by convention to include all terms invariant under $Y$-partitioning, *i.e.*, $-\sum_x^{r_x} n_x \log(n_x/N) + \log \mathcal{C}^{r_x}_N - (r_x - 1)C_{N,r_x} + C_{N,r_y^\circ}$.

Then, adopting this optimized partition for $Y$, one can apply the same dynamic programming scheme for $X$ using Eq 5 as parametric complexity and iterate the optimization of $X$ and $Y$ partitions until a stable two-state limit circle is reached. In practice, we set the initial partitioning over $X$ and $Y$ by testing equal-freq discretizations with 2 to $\lceil N^{1/3} \rceil$ bins and choosing the one which gives thehighest $I'_N(X; Y)$. We found that while the convergence speed of the iterative dynamic programming is largely independent of these initial conditions, this scheme does improve it slightly. This leads after only a few iterations to a good estimate of mutual information (averaged over limit circle) that is comparable to the existing state of the art, for both continuous and mixed-type variables, as shown below.

This optimization scheme, Eq 2, and its iterative dynamic programming computation, Eq 13, can also be adapted to compute mutual information involving joined variables, such as $I'_N(X; \{A_i\})$, with corresponding finite size correctionsand cut point encoding costs extended from Eqs 3–12. Similarly, the approach can compute conditional mutual information, such as $I'_N(X; Y|\{A_i\})$, involving continuous or mixed-type variables. To this end, $I'_N(X; Y|\{A_i\})$ needs to be defined, using the chain rule [1], as the *difference* between maximized mutual information terms involving either $\{Y, \{A_i\}\}$ and $\{A_i\}$ (Eq 14) or $\{X, \{A_i\}\}$ and $\{A_i\}$ (Eq 15) as joined

variables,

$$I'_N(X; Y|\{A_i\}) = I'_N(X; Y, \{A_i\}) - I'_N(X; \{A_i\}) \tag{14}$$

$$= I'_N(Y; X, \{A_i\}) - I'_N(Y; \{A_i\}) \tag{15}$$

Thus, starting from an initial (or optimized) partition for $X$, each term of Eq 14 is optimized with respect to $Y$ and $\{A_i\}$ partitions using Eq 4 as parametric complexity extended to multivariate categories, $n_{y,\{ai\}}$ and $n_{\{ai\}}$. Then, in turn, each term of Eq 15 is optimized with respect to $X$ and $\{A_i\}$ partitions using Eq 5 as parametric complexity extended to multivariate categories, $n_{x,\{ai\}}$ and $n_{\{ai\}}$. Note, in particular, that $\{A_i\}$ partitions are optimized *separately* for each of the four terms in Eqs 14 & 15, before taking their differences, as these optimized $\{A_i\}$ partitions might be different in general.

## Learning networks from continuous or mixed-type data

The above information maximization scheme to estimate (conditional) mutual information between continuous or mixed-type variables can then be used to extend our recent network learning algorithm MIIC [12] beyond simple categorical datasets.

**Outline of MIIC algorithm.** MIIC combines constraint-based approach and information-theoretic framework to robustly learn a broad class of causal or non-causal networks including possible latent variables [12, 13]. MIIC proceeds in three steps:

i). *Edge pruning.* Starting from a fully connected network, MIIC first removes dispensable edges by iteratively subtracting the most significant information contributions from indirect paths between each pair of variables. Significant contributors are collected based on the *3off2* score [14, 15] maximizing conditional three-point information while minimizing conditional two-point (mutual) information, which reliably assesses conditional independence, even in the presence of strongly linked variables [21]. The residual (conditional) mutual information including finite size corrections, $I'_N(X; Y|\{A_i\})$ (*i.e.* after indirect effects of significant contributors, $\{A_i\}$, have been subtracted from $I'_N(X; Y)$), is related to the removal probability of each edge, $P_{XY} = \exp(-NI'_N(X; Y|\{A_i\}))$, where $NI'_N(X; Y|\{A_i\}) > 0$ corresponds to the strength of the retained edge, as visualized by its width in MIIC graphical models [12].

ii). *Edge filtering (optional).* The remaining edges can be further filtered based on confidence ratio assessment [12], $C_{XY} = P_{XY}/\langle P^{\text{rand}}_{XY}\rangle$, where $P^{\text{rand}}_{XY}$ is the average of the probability to remove the $XY$ edge after randomly permuting the dataset for each variable. Hence, the lower $C_{XY}$, the higher the confidence on the $XY$ edge. In practice, filtering edges with $C_{XY} > 0.1$ or $0.01$ limits the false discovery rates with small datasets, while maintaining satisfactory true positive rates [12].

iii). *Edge orientation.* Retained edges are then oriented based on the signature of causality in observational data given by the sign of (conditional) three-point information [14, 15]. The final network contains up to three types of edges [12]: undirected, directed, as well as, bidirected edges, which originate from a latent variable, $L$, unobserved in the dataset but predicted to be a common cause of $X$ and $Y$, *i.e.* $X \leftarrow (L) \dashrightarrow Y$. For clarity, bidirected edges are represented with dashed lines in MIIC networks.

An important aspect of MIIC algorithm is its ability to take into account datasets with missing values, which are frequent in heterogeneous clinical datasets. In practice, MIIC computes multivariate information estimates (such as $I'_N(X; Y|\{A_i\})$) on sub-datasets for which $X$, $Y$ and

$\{A_i\}$ do not have missing values. While including iteratively additional conditioning variables $A_i$ might further restrict the size of the sub-dataset without missing value, we only consider variables $A_i$ if their missing values are missing at random (checking Kullback Leibler divergence between distributions of decreasing supports). If some data is not missing at random, the *3off2* scheme [14, 15], $I(X; Y|\{A_i\}_n) = I(X; Y) - I(X; Y; A_1) - I(X; Y; A_2|A_1) - \cdots - I(X; Y; A_n|\{A_i\}_{n-1})$, might end without finding conditional independence, ie $I(X;Y|\{A_i\}_n) > 0$, and MIIC edge pruning step is conservative by retaining the corresponding edge X-Y due to possible bias in the dataset.

MIIC's extension to continuous or mixed-type data has been implemented in MIIC online server and R package, see SI.

## Results

### Application to benchmark synthetic data

**Optimum discretization and mutual information estimates for continuous or mixed-type data.** The multivariate discretization scheme and resulting estimates of (conditional) mutual information were first benchmarked using synthetic data from known mixed or continuous probability distributions for which (conditional) mutual information can be obtained either analytically or through numerical integration. Examples of bivariate information-maximizing discretizations are shown in Fig 2 and S1 Fig for increasing sample size. The number of bins increases both with the number of samples, S1 Fig, and the magnitude of mutual information, $I_N(X; Y)$, S2A Fig. These tendencies have intuitive explanations: first, more samples means that we can assign smaller bins (width-wise) with more certainty; and second, more information means that more bins are needed to describe the interaction between the variales. We note that no single discretization of a variable $X$ can be optimal with regards to every joint distribution, see S3 Fig. While the precise cut points of variable $X$ actually depend on the

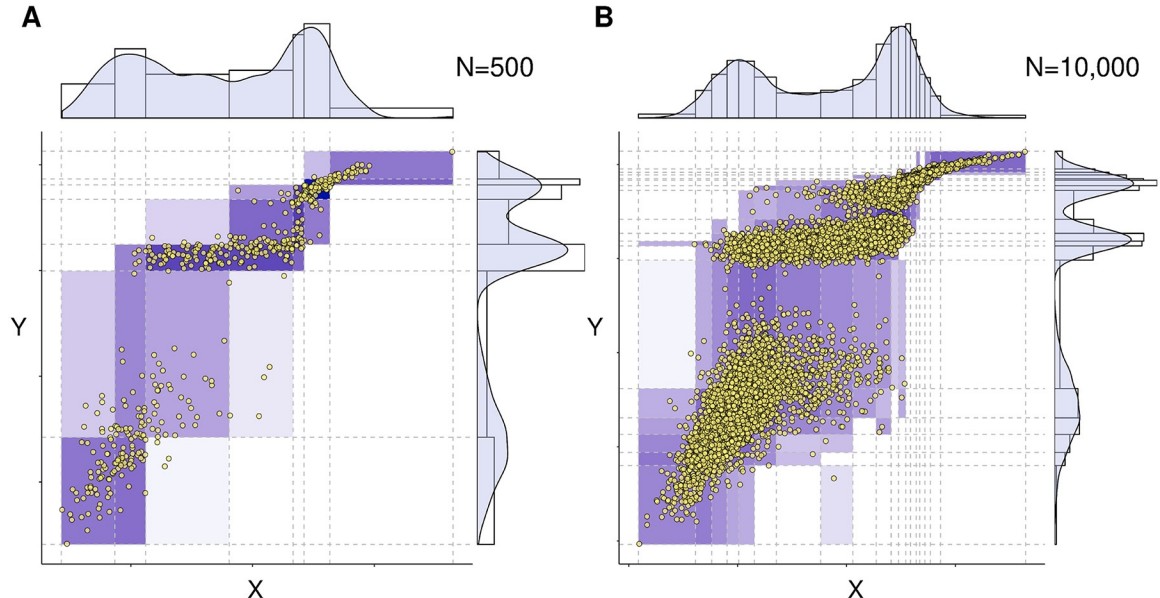

**Fig 2. Optimum bivariate discretization for mutual information estimate.** The proposed information-maximizing discretization scheme is illustrated for a joint distribution defined as a Gumbel bivariate copula with parameter $\theta = 5$ and marginal distributions chosen as Gaussian mixtures with three equiprobable peaks and respective means and variances, $\mu_X = \{0, 4, 6\}$, $\sigma_X = \{1, 2, 0.7\}$ and $\mu_Y = \{-3, 6, 9\}$, $\sigma_Y = \{2, 0.5, 0.5\}$. The information-maximizing partition yields (**A**) $I_N(X; Y) = 1.04$ for $N = 500$ samples and (**B**) $I_N(X; Y) = 1.142$ for $N = 10,000$ samples, as compared to the exact expected value $I(X; Y) = 1.205$ computed with numerical integration. See S1 Fig for additional results. Codes are provided at https://github.com/vcabeli/miic_PLoS.

variable $Y$ of interest, the number of $X$ and $Y$ bins are roughly similar(for the chosen test settings), S2A Fig, unlike found with information-maximization discretization methods lacking complexity terms [22], S2B Fig.

Next, we compared our estimation of $I_N(X; Y)$ by optimal discretization to the state of the art Kraskov–Stögbauer–Grassberger (KSG) estimator [2] for continuous distributions, specifically bivariate Gaussian distributions S4 Fig. Like otherinformation estimators based on kNN statistics, the KSG approach has a tunable parameter $k$ which will typically scale with the sample size $N$, and has to be chosen depending on the objective: the original authors recommend $k = 2$ to 4 for the best estimation, and up to $N/2$ if one is more interested in independence testing. We found that our optimal discretization with the NML complexity does indeed give a correct estimation of $I_N(X; Y)$ for all sample sizes and correlation strengths. Our approach also natively deals with categorical and mixed (*i.e.* part categorical and part continuous) variables, as the master definition of the mutual information, Eq 1, can be applied to variables of any type. Recent efforts were made to extend the KSG estimator to such cases [6–8] which are frequently encountered in real-life data, and specifically in clinical datasets. We compared the mixed-type information estimates of our method to other existing methods for varying sample sizes and found its performance to be similar or superior, S5 Fig. In addition, our information-maximizing discretization approach facilitates the interpretation of the dependences between continuous or mixed-type variables by returning their most informative categories.

Information-maximizing discretization and corresponding (conditional) mutual information estimates can be computed for any continuous or mixed-type dataset using the `discretizeMutual` function from the MIIC R package.

**Optimum discretization as an independence test between continuous or mixed-type variables.** Most importantly,our optimum discretization scheme also acts as an independence test by allowing for single bin partitions whenever no multiple-bin partitioning can glean information that is greater than its associated complexity cost. In such cases, our estimator implies variable independence, *i.e.* $I_N (X; Y) = 0$, with drastically reduced sampling error and variance, S4 Fig, as compared to other direct estimators such as KSG, which always give noisy information estimates even for vanishing mutual information between nearly independent variables and need additional hypothesis testing to be used as independence test.

Similarly, our approach robustly learns conditional independence,given a set of separating variables, $\{Z_i\}$, *i.e.*, $I_N (X; Y |\{Z_i\}) = 0$, S6 Fig, as in the case of a single common ancestor $Z$ of $X$ and $Y$, *i.e.*, $X \leftarrow Z \rightarrow Y$, with concomitant changes in optimum $X$ and $Y$ partitionings from multiple to single bins under conditioning over a continuous (S7 Fig) or categorical (S8 Fig) variable $Z$. By contrast, spurious dependency between independent variables, $X$ and $Y$, can be induced, as expected [23], by conditioning over a common descendent $Z$, as in the case of a "v-structure", $X \rightarrow Z \leftarrow Y$, S9 Fig.

Hence, the intrinsic robustness of the present optimum discretization scheme in inferring (conditional) independence and dependency is an important feature of the method as compared to kNN (conditional) information estimates, whose statistical significance remains difficult to assess in practice [2, 9].

**Reconstruction of benchmark graphical models.** We first tested the mixed-type data extension of MIIC network reconstruction method on benchmark mixed-type data. Datasets were generated based on non-linear bayesian rules using the R script provided as Supplementary code; an example of non-Gaussian mixed-type distribution dataset is shown in S10 Fig. The resulting reconstructed network F-scores are shown in Fig 3 for an increasing proportion of continuous variables over discrete variables and compared to the recent alternative methods, CausalMGM [24] and MXM [25], also designed to analyze mixed-type data. Precision, Recall and F-scores are shown for both skeleton and CPDAG in S11 and S12 Figs, respectively.

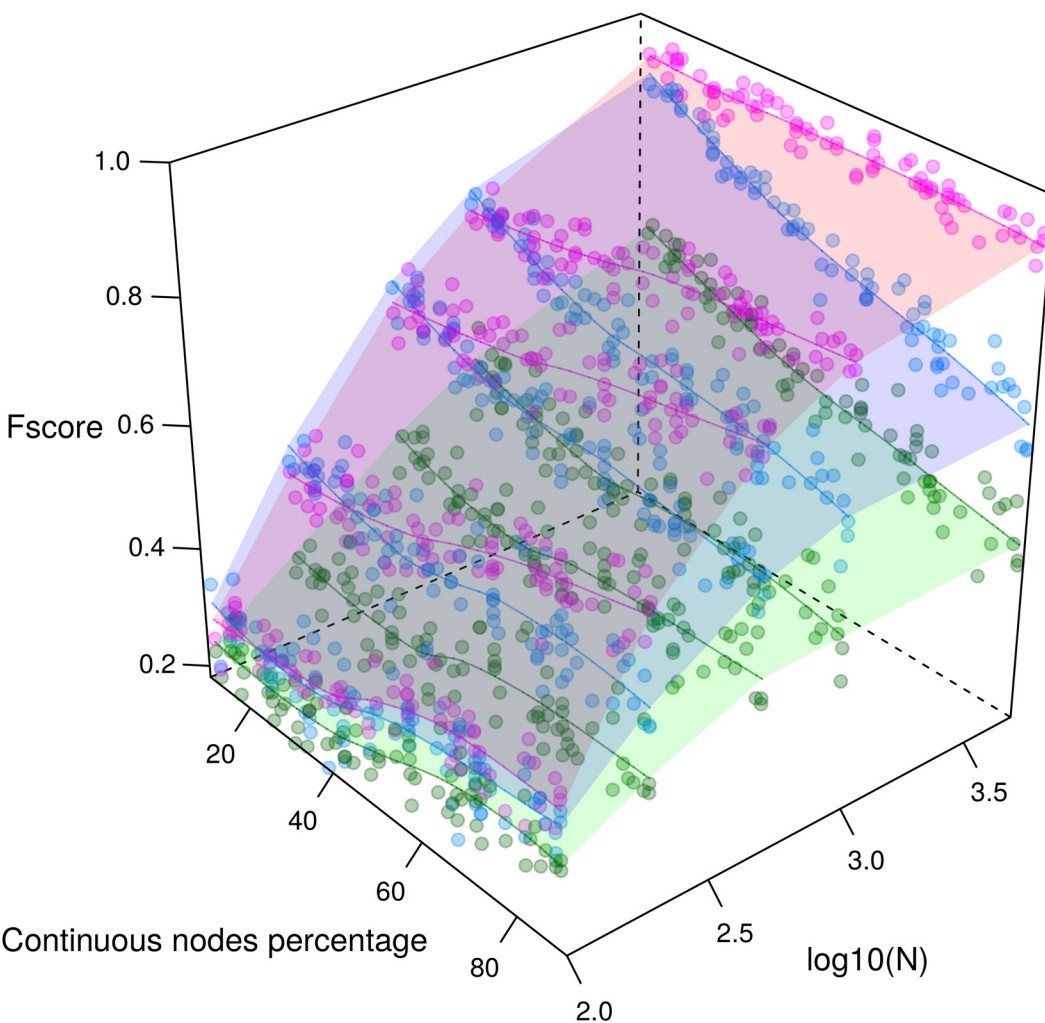

**Fig 3. Reconstruction of benchmark networks for mixed-type, non-linear, non-Gaussian datasets.** CPDAG F-scores obtained for benchmark random networks with 100 nodes and average degree 3 reconstructed from $N = 100$–5,000 samples (see histogram example S10 Fig). F-scores obtained with our parameter-free information-theoretic approach MIIC (magenta, upper surface) are compared to the best results obtained with alternative mixed-type data methods, CausalMGM [24] (blue, middle surface) and MXM [25] (green, lower surface), by optimizing CausalMGM regularization parameters ($\lambda$) and MXM significance parameter ($\alpha$), for each sample size $N$. See additional results in S11–S15 Figs. Codes are provided at https://github.com/vcabeli/miic_PLoS.

Comparisons with fully continuous datasets, S13 Fig, were also performed with additional methods, CAM [26], kPC, rank-PC and rank-FCI [27] algorithms, S14 and S15 Figs, and confirm the better performance of MIIC over alternative continuous or mixed-type network learning methods.

## Application to medical records of eldery patients with cognitive disorders

We applied this information maximization analysis for mixed-type data to reconstruct a clinical network from the medical records of 1,628 eldery patients consulting for cognitive disorders at La Pitié-Salpêtrière hospital, Paris. The dataset,provided as S1 Table, contains 107 variables of different types (namely, 19 continuous and 88 categorical variables) and heterogeneous nature (*i.e.*, variables related to previous medical history, comorbidities and

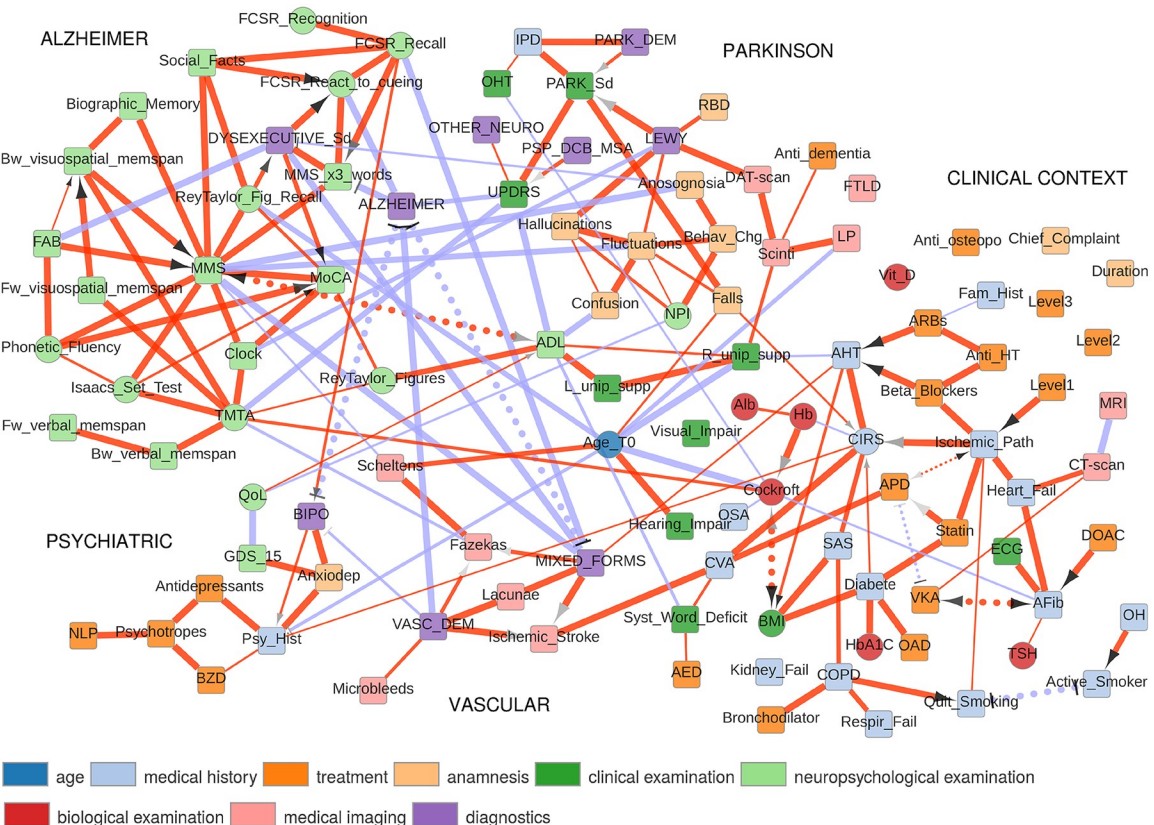

**Fig 4. Network reconstructed from medical records of 1,628 eldery patients with cognitive disorders.** Square (resp. circle) nodes correspond to discrete (resp. continuous) variables. Red (resp. blue) edges correspond to correlation (resp. anticorrelation) between variables. Dotted edges reflect latent variables, see Discussion.

comedications, scores from cognitive tests, clinical, biological or radiological examinations, diagnostics and treatments). Beyond the different types and heterogeneous nature of the recorded data, nodes of the clinical network, Fig 4, can be partitioned into groups associated to specific dementia disorders and patient clinical context, including comorbidities (diabetes, hypertension, etc) and related comedications.

**Parkisonian syndromes.**   The first group of nodes contains variables classically linked to primary degenerative dementias associated to parkinsonian syndromes (Park_Sd), notably the rarity and slowness of movements, tremor at rest and muscle stiffness, caused either by a parkinsonian dementia (PARK_DEM, 80% of cases) or a dementia with Lewy bodies (LEWY, 15% of cases). Park_Sd are identified with the Unified Parkinson Disease Rating Scale (UPDRS) which distinguishes them from Parkinson plus syndromes such as Progressive Supranuclear Palsy (PSP), Cortico Basal Degeneration (CBD) or Multiple System Atrophy (MSA). Parkinsonian syndromes are also linked to more frequent falls, idiopathic Parkinson's disease (IPD) and associated to orthostatic hypotension (OHT), in agreement with previous studies [28]. By contrast, dementia with Lewy bodies (LEWY) is found to be directly associated to cognitive fluctuations, hallucinations and Rapid eye movement sleep Behavior Disorder (RBD) as well as indirectly connected (2nd neighbor) to confusions and behavioural changes assessed through the Neuro Psychiatric Inventory (NPI) score and with a deficit of self-awareness (Anosognosia). LEWY diagnoses are also correctly associated with dopamine transporter imaging (DAT-scan) examination [29].

**Alzheimer's versus dysexecutive syndromes.**   The second and largest group of nodes mostly consists of the results from neuropsychologic tests used to assess the cognitive

functions of patients and diagnose Alzheimer's disease *versus* dysexecutive syndromes. Two types of tests can be distinguished: simple tests probing a precise cerebral function and composite tests combining the results of multiple simple tests to explore more global cognitive processes. The Trail Making Test part A (TMTA) is a simple test primarily used to examine cognitive processing speed (continuous score) by recording the time needed by the patient to connect ordered nodes (from 1 to 25) randomly placed on a sheet of paper. Our network analysis shows that TMTA is directly connected to a number of other simple tests, such as forward memory spans probing attentional capacity, backward memory spans probing immediate working memory, immediate recall of Taylor or Rey complex figures, verbal semantic fluency (Issacs set test) and the clock-drawing test. This highlights the rationale of neuropsychology in combining simple tests into more informative composite tests. Three composite tests are included in the clinical network, the Mini Mental State (MMS), the Frontal Assessment Battery (FAB) and the Montreal Cognitive Association (MoCA) tests.

- The Mini Mental State (MMS) test assesses cognitive functions related to memory, spacial and temporal orientations but not to executive functions, which require to integrate multiple information sources. MMS is found to be the main hub (with 15 neighbors) of the reconstructed network, as it is directly connected, as expected, to most of the memory test results (forward/backward verbal and visuospatial memory spans, biographic memory and delayed recalls of Taylor or Rey–Osterrieth complex figures). By constrast, MMS is found to be negatively correlated to the Alzheimer's diagnostic, through the MMS 3 word memory test, which is known to be one of the most specific tests for Alzheimer's disease, together with the Free and Cued Selective Reminding (FCSR) test. Interestingly, our network analysis shows that the Alzheimer's disease diagnostic is directly connected to the FCSR test through the low percent reactivity to cueing, which identifies genuine storage deficits (not facilitated by cueing) due to amnesic syndrome of the hippocampal type known to be characteristic of Alzheimer's disease [30].

- The Frontal Assessment Battery (FAB) test is complementary to MMS, as it is entirely focussed on executive functions, centralized in the frontal cortex; it is thus very consistent that FAB is found to be directly connected and negatively correlated to dysexecutive syndrome. Note, however, that patients suffering from dysexecutive syndrome do not typically show poor FCSR scores unlike Alzheimer patients. This confirms the specificity and sensibility of the FCSR test to Alzheimer's disease [31].

- Finally, the Montreal Cognitive Association (MoCA) composite test integrates a variety of other tests such as the clock-drawing test, the phonetic fluency test as well as semantic fluency test (Isaacs Set Test), which is consistent with the direct connections recovered between MoCA and these three individual tests in the inferred network.

**Psychiatric conditions.** The third group of nodes concerns variables associated with the psychiatric conditions of patients. It includes their past psychiatric history (Psy_Hist) and present psychiatric conditions, *i.e.*, anxio-depressive or bipolar (BIPO) syndromes, associated treatments (antidepressants, psychotropes, benzodiazepine BZD and neuroleptics NLP) and finally scores used to diagnose depression (GDS_15) and a deterioration in the quality of life (QoL). The analysis of all the links between these variables confirms the overall consistency of this psychiatric cluster: a good quality of life is closely associated with a low GDS_15 score (corresponding to a low probability of depression). Note, however, that psychiatric pathologies are all linked to each other, underlying the difficulty to distinguish them accurately. Yet, our

network analysis shows that patients with bipolar syndrome (BIPO) tend to show better scores at the FCSR recall test.

**Vascular versus mixed forms of dementias.**   The fourth group of nodes of the clinical network is associated with variables implicated in vascular dementias (VASC_DEM) originating from cerebral vascular accidents (CVA) which damage brain regions essential for cognitive processes. Different types and sizes of vascular accidents are distinguished from microbleeds to ischemic stroke (clot) and lacunae (empty spaces in the deep brain structures). These more severe vascular accidents may also lead to degenerative dementia syndromes, corresponding to a mixed form of dementia (MIXED_FORMS), which is inferred to be directly associated to low MMS scores and poor scores at the FCSR Recall test (*i.e.*, negative direct links). VASC_-DEM and MIXED_FORMS are also found to be connected to the Fazekas scale [32], which detects and quantifies white matter hyperintensities in the brain that are the consequence of cerebral small vessel disease including demyelination and axonal loss of neuronal cells. The Fazekas scale is found to be directly associated to low cognitive processing speed (TMTA) and also strongly correlated to the Scheltens scale [33] quantifying the severity of hippocampal atrophy, in agreement with a recent independent report [34]. The hippocampus is a brain structure involved in memory and space navigation, which is consistent with our finding of a direct negative association between Scheltens scale and MMS score. Interestingly, this predicted association between the Fazekas and the Scheltens scales, inferred from our unsupervised global network analysis, provides some physiological insights linking the consequence of vascular accidents (Fazekas scale) to the atrophy of important brain structures (Scheltens scale) and, thereby, to cognitive and functional impairments, as reported in clinical studies linking white matter hyperintensities (Fazekas scale) to cognitive impairment [35].

**Patient clinical context.**   The last important group of nodes of the clinical network includes variables associated with the patient clinical context including comorbidities, related examinations and treatments. These are different anterior chronic diseases, such as arterial hypertension (AHT), diabetes, chronic obstructive pulmonary disease (COPD), atrial fibrillation (AFib), that might have an impact on the patient's vital prognosis. All the links within this comorbidity cluster are very consistent, each pathology being directly associated with its known risk and predisposition factors, biological markers, specific examinations and treatments. In particular, diabetes is associated with a high body mass index (BMI), glycated hemoglobin blood test (HbA1c), treatment by oral antidiabetic (OAD) drugs and statin; COPD is associated with sleep apnea syndrome (SAS) and the risk of respiratory failure, the use of bronchiodilator drugs and the necessity to quit smoking; AHT is associated with an increase risk of mixed form dementia and treatments by angiotensin receptor blockers (ARBs), beta-blockers and other anti-hypertension (Anti HT) drugs; Finally, AFib, detected by electrocardiogram (ECG), is associated with an increased risk of heart failure and high levels of thyroid-stimulating hormone (TSH) and treated with vitamine K antagonist (VKA) and direct oral anticoagulants (DOAC).

## Discussion

We report in this paper a novel optimal discretization method to simultaneously compute and assess the significance of mutual information, as well as conditional multivariate information, between any combination of continuous or mixed-type variables. The approach is used to reconstruct graphical models from mixed-type datasets by uncovering direct, indirect and possibly causal relationships in complex heterogenous data. The method is shown to outperform state-of-the-art approaches on benchmark mixed-type datasets, before being applied to analyze the medical records of eldery patients with cognitive disorders from La Pitié-Salpêtrière Hospital, Paris.

From a methodological perspective, this information-maximizing discretization approach facilitates the interpretation of either the dependences or the independencies between continuous or mixed-type variables. First, obtaining optimal discretization helps explain the dependences in terms of the most informative categories of continuous variables. Second, and most importantly, optimal discretization also acts as an independence test by allowing for single bin partitions whenever multiple-bin partitioning provides less information than its associated complexity cost.

From the perspective of clinical applications, the method is able to globally uncover interdependences within complex heterogeneous data from medical records without specific hypothesis nor prior knowledge on any clinically relevant information. The reconstructed clinical network from cognitive disorder patients (Fig 4) recovers well known as well as novel direct and indirect relations between medically relevant variables.

In addition, we found that this reconstructed clinical network captures also some facets of the neurologist's reasoning behind the diagnoses of distinct dementias. In particular, diagnosis nodes can be interpreted as "explanatory" variables associated to a number of "explaining-away effects" [23] in the form of "v-structures", *i.e.*, $D_1 \rightarrow S/E \leftarrow D_2$, whenever alternative diagnoses, $D_1$ or $D_2$, can independently explain a given syndrome, $S$, or the result of a specific examination, $E$. Examples discussed in more details above are PARK_DEM $\rightarrow$ PARK_Sd $\leftarrow$ LEWY, VASC_DEM $\rightarrow$ Fazekas $\leftarrow$ MIXED_FORMS and VASC_DEM $\rightarrow$ Ischemic_Stroke $\leftarrow$ MIXED_FORMS. In addition, anticorrelations between different diagnostic nodes reflect the alternative choices of diagnosis by the neurologist, either in the form of "differential diagnoses" through a reasoning by elimination, in particular, to diagnose Alzheimer's disease, *i.e.*, VASC_DEM ⊣ ALZHEIMER, or in the form of a latent variable, visualized as bidirected dotted edges and corresponding to alternative diagnoses by the neurologist, *i.e.*, ALZHEIMER←--*diagnosis*--→MIXED_FORMS or ALZHEIMER←--*diagnosis*--→BIPO. Latent variables may also represent the clinician's decisions between alternative treatments, *e.g.*, APD←--*clinician_decision*--→VKA or a nonrecorded or implicit information in the patient personal or medical history, *e.g.*, active_smoker←--*ever_smoked*--→quit_smoking, Fig 4.

The main strengths of our clinical network reconstruction method are three-fold. First, it performs an unbiased check on the database content (expected, yet missing direct links in the reconstructed network hint to likely problems in the database *e.g.*, erroneous or missing data). Second, it does not need any expert-informed hypothesis and provides, without prior knowledge in the field, graphical models complementing analyses by experts. Finally, it can discover novel unexpected direct interdependencies between clinically relevant information, such as the direct connection between Fazekas and Scheltens scales, Fig 4, which may provide some physiological insights and suggest new research directions for further investigation.

Hence, beyond the challenge of learning clinical networks from mixed-type data, our method offers a user-friendly global visualisation tool of complex, heterogeneous clinical data which could help other practitioners visualize and analyze direct, indirect and possibly causal effects from patient medical records.

## Supporting information

**S1 File. Supplementary Materials and Methods.** Benchmark data generation (continuous and discrete variables). Performance measures. Benchmark parameter tuning. Resource availability.
(PDF)

**S1 Table. Dataset from 1,628 eldery patients with cognitive disorders from La Pitié-Salpê-trière hospital, Paris.** The dataset, fully deidentified, contains 107 variables of different types

(namely, 19 continuous and 88 categorical variables) and heterogeneous nature (*i.e.*, variables related to previous medical history, comorbidities and comedications, scores from cognitive tests, clinical, biological or radiological examinations, diagnostics and treatments).
(XLSX)

**S1 Fig. Optimum bivariate discretization for mutual information estimation.** The proposed information-maximizing discretization scheme is illustrated for a joint distribution defined as a Gumbel bivariate copula with parameter $\theta = 5$ and univariate marginal-distribution functions chosen as Gaussian mixtures with three equiprobable peaks and respective means and variances, $\mu_X = \{0, 4, 6\}$, $\sigma_X = \{1, 2, 0.7\}$ and $\mu_Y = \{-3, 6, 9\}$, $\sigma_Y = \{2, 0.5, 0.5\}$. Information-maximizing partitions are displayed for different sample sizes with corresponding mutual information estimates: **(A)** $N = 100$ samples, $I_N(X; Y) = 0.928$ (and $I'_N(X; Y) = 0.649$); **(B)** $N = 500$ samples, $I_N(X; Y) = 1.040$ (and $I'_N(X; Y) = 0.866$); **(C)** $N = 1,000$ samples, $I_N(X; Y) = 1.096$ (and $I'_N(X; Y) = 0.977$); **(D)** $N = 10,000$ samples, $I_N(X; Y) = 1.142$ (and $I'_N(X; Y) = 1.075$). The actual mutual information value was computed through numerical integrationof the marginals and the joint probability distribution and yields, $I(X; Y) = 1.205$, in good agreement with the obtained estimates for large $N$.
(EPS)

**S2 Fig. Adaptive information-maximizing partitions depending on interaction strength.** To assess the range in bin numbers depending on the strength of interaction between variables, we generated $N = 1,000$ independent samples for 10,000 Gaussian bivariate distributions with a uniformly distributed correlation coefficient $\rho$ in $[-1, 1]$. The real mutual information (*RI*) of Gaussian bivariate distributions can be computed directly [1], as $RI(X; Y) = -\log(1 - \rho^2)/2$. For each pair $(X, Y)$, we estimated the mutual information with the proposed optimum bivariate discretization as well as the Maximal Information Coefficient [22] using the `minepy` package [36] **(A)** The information-maximizing partition proposed in the present paper behaves as expected: the number of bins on each variable is roughly similar and scales monotonically with the strength of the interaction between variables. This implies that additional bins are only introduced when their associated complexity cost is justified by a larger gain in mutual information. Conversely, when the information between $X$ and $Y$ approaches zero, both variables are partitioned into fewer and fewer bins until a single bin is selected for each variable, when they are inferred to be independent, given the available data. **(B)** The partition chosen to estimate the Maximal Information Coefficient is very different, regardless of the interaction strength, as it systematically corresponds to an unbalanced distribution of bins between the two variables, with one variable usually partitioned into the maximum number of bins(set by default to floor($N^{0.6}/2$) = 31) while the other is discretized into two levels only. This result is not unexpected, however, as the Maximal Information Coefficient [22] is defined by maximizing the mutual information of the discretized variables over the grid, $I([X]_{\Delta_x}; [Y]_{\Delta_y})$, *normalized by the minimum* of $\log \Delta_x$ and $\log \Delta_y$. Indeed, maximizing the normalized mutual information is done by partitioning as few samples as possible into the maximum number of bins in one dimension (as sketched in Fig 1), while simultaneously minimizing the number of bins, and thus $\log \Delta_i$, in the other dimension. See further discussion in [37].
(EPS)

**S3 Fig. Interaction-dependent optimum discretization.** Optimum bivariate partitions obtained from $N = 1,000$ samples of two different joint distributions $P(X, Y)$ sharing the same sampling of $X$ taken from a uniform distribution on $[0, 0.3]$, but with different dependences for $Y$. **(A)** $Y$ is defined as $\log(X) + \epsilon_1$, and **(B)** $Y$ is defined as $X^5 + \epsilon_2$, where $\epsilon_1$ and $\epsilon_2$ are Gaussian noise terms chosen so that the mutual informations of both examples are

comparable, $I(X;Y) \simeq 0.75$. This example shows that the optimum partition for $X$ depends on its specific relation with $Y$ and needs to be discretized with finer partitions in **(A)** at low $X$ values for which $Y \simeq \log X$ varies the most and in **(B)** at higher $X$ values for $Y \simeq X^5$.
(EPS)

**S4 Fig. Mutual information estimation for Gaussian bivariate distributions.** 100 bivariate normal distributions were sampled for varying sample sizes, increasing from top to bottom, and correlation coefficients $\rho$ ranging from 0.01 to 0.9. The mutual information was estimated with the proposed optimum discretization scheme and the KSG estimator with different parameters $k$. The mean squared error (center graphs) was calculated thanks to the analytical result of the mutual information of the bivariate Gaussian: $I(X; Y) = -\log(1 - \rho^2)/2$. The standard deviation of each estimator over the 100 replications was also plotted against the correlation coefficient (right).
(EPS)

**S5 Fig. Mutual information estimation of mixed variables.** Experiment set-ups and analytical values for the mutual information were taken fom [7] and 50 runs were performed for each sample size $N$. Our proposed approach is compared to a naive equal-frequency discretization with $N^{1/3}$ bins, a kernel and a noisy KSG estimator as implemented in JIDT [38], as well as the recent KSG extensions for estimating the mutual informmation between a categorical and a continuous variable (mixed KSG Ross [6]), and between mixed-type variables (mixed KSG Gao [7]). For all nearest-neighbour based approaches, the number of nearest neighbours was set to $k = 5$. From left to right, top to bottom, the simulations are devised after experiment I, experiment II, experiment IV with $p = 0$ and experiment IV with $p = 0.15$, from [7].
(EPS)

**S6 Fig. Conditional mutual information estimation for multivariate Gaussian distributions.** Four-dimensional normal distributions $P(X, Y, Z_1, Z_2)$ were sampled for $N = 100$ to 5, 000 samples 100 times for each correlation coefficient $\rho = \rho_{XY}$, chosen between 0.05 and 0.95. The other pairwise correlation coefficients were fixed as $\rho_{XZ_1} = \rho_{XZ_2} = \rho_{YZ_1} = \rho_{YZ_2} = \lambda = 0.7$ and $\rho_{Z_1 Z_2} = 0.9$. The conditional mutual information $I(X; Y|Z_1, Z_2)$ was then estimated using the proposed optimum partitioning scheme as well aswith kNN conditional information estimates as in S4 Fig. $\rho$ values closed to zero, mimick "V-structures" as they correspond to pairwise independence but conditional dependence; by constrast $\rho = 2\lambda^2/(1 + \rho_{Z_1 Z_2}) \simeq 0.5158$ corresponds to conditional independence, while $\rho > 0.5158$ impliesthat $X$ and $Y$ share more information than the indirect flow through $Z_1$ and $Z_2$. The analytical value of the conditional mutual information is derived as follows; given the $4 \times 4$ covariance matrix $\Sigma$ and its four $2 \times 2$ partitions $\Sigma_{ij}$, we first compute the conditional covariance matrix $\bar{\Sigma} = \Sigma_{11} - \Sigma_{12}\Sigma_{22}^{-1}\Sigma_{21}$ where $\Sigma_{22}^{-1}$ is the generalized inverse of $\Sigma_{22}$. The partial correlation between $X$ and $Y$ is obtained as $\rho_{XY \cdot Z_1 Z_2} = \bar{\Sigma}_{12}/\sqrt{\bar{\Sigma}_{11} * \bar{\Sigma}_{22}}$, and the analytical conditional mutual information for a multivariate normal distribution is given by $I(X; Y|Z_1, Z_2) = -\log(1 - \rho^2_{XY \cdot Z_1 Z_2})/2$.
(EPS)

**S7 Fig. Pairwise dependence and conditional independence between $X$ and $Y$ sharing a common cause $Z$.** This example illustrates the (conditional) correlation patterns emerging from the presence of a confounding variable, as depicted by the causal diagram $X \leftarrow Z \rightarrow Y$. $Z$ is generated with a uniform law $U(0, 1)$ for $N = 1, 000$ observations and $X, Y$ are both defined as $2Z + \epsilon$ with independent normal noise $\epsilon \sim \mathcal{N}(0, 0.2)$. **(A)** optimum discretization maximizing $I'_N(X; Y)$ with a strong pairwise correlation, and **(B)** optimum discretization which maximizes the conditional mutual information with finite size correction, $I'_N(X; Y|Z)$. In the latter

case, the optimum discretization scheme results in a single bin on both variables as the flow information between $X$ and $Y$ is blocked by conditioning on the common cause $Z$.
(EPS)

**S8 Fig. Pairwise dependence and conditional independence between non Gaussian $X$ and $Y$ sharing a common categorical cause.** Another confounding example, $X \leftarrow Z \rightarrow Y$, taken from [25] with a uniform categorical $Z$ with three levels, $X$ and $Y$ being continuous, for $N = 1,000$ observations. With $Z_i$ the binary variable corresponding to the $i$-th dummy variable of $Z$, we defined $X = -Z_1 + Z_2 + 0.2\epsilon_X$ which is centered around either $-1$ if $Z = 1$, 0 if $Z = 3$ or 1 if $Z = 2$; and $Y = Z_1 + Z_2 + 0.2\epsilon_Y$, $\epsilon \sim \mathcal{N}(0, 1)$ which is centered around either 0 if $Z = 3$ or 1 if $Z = 1$ or $Z = 2$. As for continuous common cause in S7 Fig, there is **(A)** some non-zero mutual information between $X$ and $Y$ corresponding to an optimum discretization, while **(B)** conditional mutual information vanishes when conditioning on the categorial common cause, $Z$, with the partitions of both $X$ and $Y$ variables consisting in a single bin.
(EPS)

**S9 Fig. Pairwise independence and conditional dependence with a v-structure.** Example of two independent variables $X$, $Y$ both causing a third variable $Z$ as: $X \rightarrow Z \leftarrow Y$. $N = 1,000$ observations are drawn for $X, Y \sim \mathcal{N}(0, 1)$ and $Z = X + Y$. **(A)** The two variables $X$ and $Y$ being independent, no multi-bin discretization can be found to yield an information estimate that is greater than the corresponding complexity cost. However, **(B)** conditioning on the common effect $Z$ 'activates' the v-structure path generating a spurious relationship between $X$ and $Y$. This is reflected in the fact that the induced interaction between $X$ and $Y$ requires a multiple bin optimum discretization to estimate $I_N(X; Y|Z) = 1.188$ (with $I'_N(X; Y|Z) = 0.745$).
(EPS)

**S10 Fig. Example of dataset generated for mixed-type, non-linear, non-Gaussian benchmarking with 69 continuous and 31 categorical variables.** Each plot represents the observed density or histogram ($N = 1,000$) of the continuous or categorical variable $X_i$, constructed by structural equation models given its parents' distributions (see Supporting Information).
(EPS)

**S11 Fig. Skeleton assessment of benchmark networks for mixed-type, non-linear, non-Gaussian datasets.** Skeleton Precision, Recall and F-scores obtained for benchmark random networks with 100 nodes and average degree 3 reconstructed from $N = 100$–5,000 samples (see histogram example Fig. S11). Performances obtained with our parameter-free information-theoretic approach MIIC (magenta) are compared to the results obtained with the best parameterization (maximizing the skeleton F-score) of CausalMGM [24] (blue) and MXM [25] (green). See Supporting Information.
(EPS)

**S12 Fig. CPDAG assessment of benchmark networks for mixed-type, non-linear, non-Gaussian datasets.** CPDAG Precision, Recall and F-scores obtained for benchmark random networks with 100 nodes and average degree 3 reconstructed from $N = 100$–5,000 samples (see histogram example S11 Fig). Performances obtained with our parameter-free information-theoretic approach MIIC (magenta) are compared to the results obtained with the best parameterization (maximizing the CPDAG F-score) of CausalMGM [24] (blue) and MXM [25] (green). See Supporting Information.
(EPS)

**S13 Fig. Example of dataset used for continuous, non-linear, non-Gaussian benchmarking with 100 continuous variables.**
(EPS)

**S14 Fig. Skeleton assessment of benchmark networks for continuous, non-linear, non-Gaussian datasets.** Skeleton Precision, Recall and F-scores obtained for benchmark random networks with 100 nodes and average degree 3 reconstructed from $N = 100 - 10,000$ samples (see histogram example Fig. S14). Results obtained with our parameter-free information-theoretic approach MIIC are compared for optimum non-uniform bin sizes and for equal frequency bin sizes (with $N^{1/3}$ bins) as well as to the best results obtained with alternative continuous data methods: PC with Gaussian conditional independence test, rankPC and rankFCI from the `pcalg` package [27], kPC with theHelbert-Schmidt Independence Criterion [39, 40] and CAM [26] algorithms, after optimizing their respective parameter ($\alpha$) for each sample size $N$. See Supporting Information.
(EPS)

**S15 Fig. CPDAG assessment of benchmark networks for continuous, non-linear, non-Gaussian datasets.** CPDAG Precision, Recall and F-scores obtained for benchmark random networks with 100 nodes and average degree 3 reconstructed from $N = 100 - 10,000$ samples (same simulation settings as in Fig. S15).
(EPS)

## Acknowledgments

We thank Etienne Birmelé, Pierre Charbord, Eric Gaussier, Gregory Nuel, Elisabeth Remy, Denis Thieffry for discussions.

## Author Contributions

**Conceptualization:** Hervé Isambert.

**Data curation:** Louis Verny, Nadir Sella, Marc Verny.

**Formal analysis:** Hervé Isambert.

**Funding acquisition:** Hervé Isambert.

**Investigation:** Vincent Cabeli, Louis Verny, Nadir Sella, Guido Uguzzoni, Marc Verny, Hervé Isambert.

**Methodology:** Vincent Cabeli, Guido Uguzzoni, Hervé Isambert.

**Project administration:** Hervé Isambert.

**Resources:** Vincent Cabeli, Louis Verny, Nadir Sella, Marc Verny.

**Software:** Vincent Cabeli, Nadir Sella.

**Supervision:** Marc Verny, Hervé Isambert.

**Validation:** Vincent Cabeli, Louis Verny, Guido Uguzzoni, Marc Verny.

**Visualization:** Vincent Cabeli, Nadir Sella.

**Writing – original draft:** Vincent Cabeli, Louis Verny, Hervé Isambert.

**Writing – review & editing:** Hervé Isambert.

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
