## [Decision Letter · Decision Letter 0]

21 Nov 2019

Dear Dr Isambert,

Thank you very much for submitting your manuscript 'Learning clinical networks from medical records based on information estimates in mixed-type data' for review by PLOS Computational Biology. Your manuscript has been fully evaluated by the PLOS Computational Biology editorial team and in this case also by independent peer reviewers.

The reviewers found the presented work to be timely and interesting. However, they raised some substantial concerns about the manuscript as it currently stands. In particular, the manuscript needs to be substantially revised to improve the clarity and usability and also how this work compares to the existing work. The manuscript needs to provide a proper Methods section. The code also needs to be made available along with example inputs and outputs. 

While your manuscript cannot be accepted in its present form, we are willing to consider a revised version in which the issues raised by the reviewers have been adequately addressed. We cannot, of course, promise publication at that time.

Sincerely,

Sushmita Roy, Ph.D.

Associate Editor

PLOS Computational Biology

Mark Alber

Deputy Editor

PLOS Computational Biology

[LINK]

Reviewer's Responses to Questions

**Comments to the Authors:**

Reviewer #1: The authors present a method for computing the mutual information between mixed variables by finding an optimal binning strategy. They demonstrate that the method is competitive with state-of-the-art methods for estimating mutual information between mixed variables and that it has a particular advantage as an independence test. They then apply this mutual information estimator to graphical model structure learning and demonstrate good performance on benchmark data as well as present a case study application to a medical data.

The method and application are technically sound and well-presented.

My two main concerns are:

1. In the author summary and introduction an impression is built up that no methods exist for computing mutual information for mixed variables. The authors are clearly aware of these methods (references 15-17), however, the mention of these methods is pushed down deep into the benchmarking subsection of the results section. These must be brought to the forefront (be referenced in the introduction) as not to misrepresent the state of the art.

2. There's no explanation of the principle by which "latent variables" are suggested in the graphical model, i.e. what makes an edge suggest mediation by a latent variable vs a simple correlation/anticorrelation edge. If this is a post-hoc decision in light of expert knowledge the text needs to be explicit about that.

Reviewer #2: Review of the PLOS Computational Biology manuscript #PCOMPBIOL-D-19-01535 "Learning clinical networks from medical records based on information estimates in mixed-type data" by V Cabeli, L Verny, N Sella, G Uguzzoni, M Verny, H Isambert

Summary:

This paper presents an extension of the MIIC network learning algorithm for mixed-type (i.e. both continuous and categorical) data. This new approach relies on a new estimation procedure for the (conditional) Mutual Information (MI) for such mixed-type data, also introduced in this manuscript. After introducing the need and relevance of such methods especially in the context of medical records, the authors present new methodological developments for estimating (conditional) MI, that is suitable for mixed-type data, and illustrate its good performance on benchmark synthetic. Then the authors outline their extension of the MIIC algorithm for mixed data, briefly benchmark it, and present an extensive application to medical records of elderly patients with cognitive disorders. Finally, a short discussion quickly highlights the conclusions from that application.

General Comments:

This manuscript presents an interesting and timely new method for estimating network from mixed-type data such as medical records. While the manuscript is well written, the structure is a bit confusing and impedes both its readability and assessment: first it lacks a materials and methods section which should contains the methodological developments that are currently being presented alongside simulations benchmarks and application in the Results section; secondly the Discussion section should be broader and better acknowledge the assumptions and limitations made by the proposed method. Besides, I have questions concerning the guarantees offered by the proposed method and the assumptions required, as those are not clearly outlined in the manuscript. In particular, I wonder how the authors deal with the scaling of the MI and how it impacts edge pruning and filtering in their network inference. My questions to the authors are detailed below.

Major issues:

1. The MI is an unbounded positive quantity, therefore one of the difficulties of using MI for inferring networks from mixed-type data is the scaling of the MI that will usually varies depending on the variable type (binary, categorical, continuous…). This aspect should be discussed in the manuscript. In particular, the MI for categorical variables tends to increase with the number of categories. How do the proposed method deals with this when i) pruning (and filtering) the edges of the inferred network ? ii) representing the association strength such as in Figure 4 ?

2. The manuscript lacks a method section. New methodological development should be in a specific Methods section, with a first subsections presenting the new approach for approximating partial MI in mixed-data and a second one presenting the extension of the MIIC algorithm.

3. The discussion section should discuss the whole scope of the manuscript, including assumptions and limitations of the proposed approach for learning network from mixed-data, as well as synthetic benchmark results and application.

4. Page 4 line 82-33, the authors seem to make an assumption on the partitioning cut-points that should be clarified, especially if it is required for their approximation to be accurate.

5. The authors should detail a bit more how they derived equation 7 or provide a reference.

6. It is unclear whether there are guarantees for the convergence of the proposed optimization procedure presented at the bottom of page 4, or if this is more of a heuristic procedure that works in practice.

7. The authors should describe what are X and Y represented on Figure 2 and how they are generated in the synthetic benchmark (this is somewhat explained in the SI but should be mentioned and clarified in the main manuscript).

8. Page 6 the authors alludes to the capacity of their approach to identify (conditional) independence. Could they clarify how do they characterize independence from (part) MI — in my experience this can be difficult in practice, even with resampling procedures?

9. I command the author in making a software available for their method in the form of the R package miic. However, I was unable to find (and so test) the mentioned discretizeMutual function neither from the CRAN version of the package or on GitHub. The authors should provide an url for the code of the proposed approach.

Minor issues:

1. Page 2 line 36, “cause-effet” should appear in English

2. Figure 1 & 3 should have a linetype/color legend and should be readable in black & white

3. Page 6 line 142 KSG acroym is never defined

**Have all data underlying the figures and results presented in the manuscript been provided?**

Reviewer #1: No: The data generation process for the benchmark data was described, which may serve as a substitute for explicit tables with that data (or the authors can easily generate such tables and provide them as, e.g. csv files).

The authors didn't specify how and whether the clinical data from the case study could be accessed (likely one of the options for sensitive data will apply).

Reviewer #2: Yes

PLOS authors have the option to publish the peer review history of their article (what does this mean?). If published, this will include your full peer review and any attached files.

Reviewer #1: No

Reviewer #2: No

---

## [Decision Letter · Decision Letter 1]

8 Mar 2020

Dear Dr. Isambert,

Thank you very much for submitting your manuscript "Learning clinical networks from medical records based on information estimates in mixed-type data" for consideration at PLOS Computational Biology. As with all papers reviewed by the journal, your manuscript was reviewed by members of the editorial board and by several independent reviewers. The reviewers appreciated the attention to an important topic. Based on the reviews, we are likely to accept this manuscript for publication, providing that you modify the manuscript according to the review recommendations.

In particular, please provide the necessary data files underlying the figures and results in your manuscript.

Sincerely,

Sushmita Roy, Ph.D.

Associate Editor

PLOS Computational Biology

Mark Alber

Deputy Editor

PLOS Computational Biology

[LINK]

Reviewer's Responses to Questions

**Comments to the Authors:**

Reviewer #1: This is a review of the revised version of PCOMPBIOL-D-19-01535R1 'Learning clinical networks from medical records based on information estimates in mixed-type data'

The authors present a method for computing the mutual information between mixed variables by finding an optimal binning strategy. They demonstrate that the method is competitive with state-of-the-art methods for estimating mutual information between mixed variables and that it has a particular advantage as an independence test. They then apply this mutual information estimator to graphical model structure learning and demonstrate good performance on benchmark data as well as present a case study application to a medical data.

The method and application are technically sound and well-presented.

The authors' revisions have fully addressed all the concerns I had in my prior review.

I recommend publication as dissemination of this method will be of value to the research community.

Reviewer #2: The authors have adequaltely adressed all my comments. I recommend that the authors pay extra attention to provide **all data underlying the figures and results presented in the manuscript** in their final submission (especially regarding Fig 2 and 4).

**Have all data underlying the figures and results presented in the manuscript been provided?**

Reviewer #1: Yes

Reviewer #2: Yes

PLOS authors have the option to publish the peer review history of their article (what does this mean?). If published, this will include your full peer review and any attached files.

Reviewer #1: Yes: Yuriy Sverchkov

Reviewer #2: No
---

## [Editor Report · Decision Letter 2]

10 Apr 2020

Dear Dr. Isambert,

We are pleased to inform you that your manuscript 'Learning clinical networks from medical records based on information estimates in mixed-type data' has been provisionally accepted for publication in PLOS Computational Biology.

Best regards,

Sushmita Roy, Ph.D.

Associate Editor

PLOS Computational Biology

Mark Alber

Deputy Editor

PLOS Computational Biology

---

## [Editor Report · Acceptance letter]

28 Apr 2020

PCOMPBIOL-D-19-01535R2 

Learning clinical networks from medical records based on information estimates in mixed-type data

Dear Dr Isambert,

I am pleased to inform you that your manuscript has been formally accepted for publication in PLOS Computational Biology. Your manuscript is now with our production department and you will be notified of the publication date in due course.

With kind regards,

Laura Mallard
